# ID and OOD Performance Are Sometimes Inversely Correlated on Real-world Datasets

**Damien Teney**
Idiap Research Institute, Switzerland
`damien.teney@idiap.ch`

**Yong Lin**
Hong Kong Uni. of Science and Technology
`ylindf@connect.ust.hk`

**Seong Joon Oh**
University of Tübingen, Germany
`coallaoh@gmail.com`

**Ehsan Abbasnejad**
University of Adelaide, Australia
`ehsan.abbasnejad@adelaide.edu.au`

## Abstract

**Context.** Several studies have compared the in-distribution (ID) and out-of-distribution (OOD) performance of models in computer vision and NLP. They report a frequent positive correlation, but surprisingly, almost never an inverse correlation that would be indicative of a necessary trade-off. Such inverse patterns are possible theoretically, and their occurrence in practice is important to determine whether ID performance can serve as a proxy for OOD generalization.

**Findings.** This paper shows that inverse correlations between ID and OOD performance do happen with multiple real-world datasets, not only in artificial worst-case settings. We explain theoretically how these cases arise and how past studies missed them because of improper methodologies that examined a biased selection of models.

**Implications.** Our observations lead to recommendations that contradict those found in much of the current literature.

- High OOD performance sometimes requires trading off ID performance.
- Focusing on ID performance alone may not lead to optimal OOD performance. It may produce diminishing (eventually negative) returns in OOD performance.
- In these cases, studies on OOD generalization that use ID performance for model selection (a common recommended practice) will necessarily miss the best-performing models, making these studies blind to a whole range of phenomena.

## 1 Introduction

**Past observations.** This paper complements existing studies that empirically compare in-distribution (ID) and out-of-distribution[1] (OOD) performance of deep learning models [2, 6, 25, 22, 24, 42, 50]. It has long been known that models applied to OOD data suffer a drop in performance, e.g. in classification accuracy. The above studies show that, despite this gap, ID and OOD performance are often positively correlated[2] across models on benchmarks in computer vision [25] and NLP [24].

**Past explanations.** Frequent positive correlations are surprising because inverse correlations are similarly possible theoretically. Indeed, ID and OOD data contain different associations between labels and features. One could imagine e.g. that an image background is associated with class $\mathcal{C}_1$ ID and class $\mathcal{C}_2$ OOD. The more a model relies on the presence of this background, the better its

---

[1]We use "OOD" to refer to test data conforming to covariate shifts [41] w.r.t. the training data.
[2]We use "correlation" to refer both to linear and non-linear relationships.

37th Conference on Neural Information Processing Systems (NeurIPS 2023).

Figure 1: Several past studies suggest that positive correlations between ID / OOD performance are ubiquitous. This paper shows empirically and theoretically why inverse correlations are also possible and can be accidentally overlooked. The possibility of ID / OOD trade-offs goes counter the common practice of model selection based on ID performance, which is recommended in many benchmarks for OOD generalization.

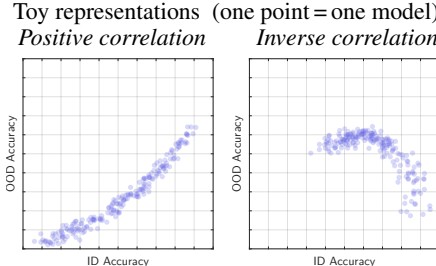

Toy representations  (one point = one model)

*Positive correlation*      *Inverse correlation*

ID performance but the worse its OOD performance, resulting in an inverse correlation. The fact that inverse correlations were almost never observed in practice was explained with the fact that **real-world benchmarks only contain mild distribution shifts** [22]. However, we will show in Section 4 that the reason may also be in flawed experimental design.

A recent study [50] shows various correlation patterns across datasets, but never *inverse* ones:

> "*We did not observe any trade-off between accuracy and robustness, where more accurate models would overfit to spurious features that do not generalize.*" [50]

This does not match our past general experience, so we set out to expose in this paper actual cases of inverse correlations, using popular datasets used for OOD research. Our results show inverse correlations across models trained with varying numbers of epochs and random seeds. These patterns are even more striking when models are trained with a regularizer that identifies a diverse set of solutions to the ERM objective [46].

**Explaining inverse correlations.** We name the underlying cause a "*misspecification*", an extension of the "*underspecification*" used previously to explain why models with similar ID performance can vary in OOD performance [5, 16, 47]. In cases of misspecification, the standard ERM objective (empirical risk minimization) aligns with a maximization of the ID performance but is in conflict with the OOD performance. ID and OOD metrics then vary inversely to one another. In Appendix C, we present a minimal theoretical example to illustrate how inverse correlations originate from the presence of both robust and spurious features in the data. In Section 7, we show that different patterns of ID / OOD performance occur with different magnitudes of distribution shifts.

**Summary of contributions.**
- An empirical examination of ID vs. OOD performance on several real-world datasets showing inverse correlation patterns that conflict with past evidence (Section 3).
- An explanation and an empirical verification of why past studies missed such patterns (Section 4).
- A theoretical analysis showing how inverse correlation patterns can occur (Appendix C).
- A revision of conclusions and recommendations made in past studies (Section 8).

## 2  Previously-observed patterns of ID / OOD performance

Multiple studies conclude that ID and OOD performance vary jointly across models on many real-world datasets [6, 25, 42]. Miller et al. [25] report an almost-systematic linear correlation[3] between probit-scaled ID and OOD accuracies. Mania and Sra [22] explain this trend with the fact that many benchmarks contain only mild shifts.[4] Andreassen et al. [2] find that pretrained models perform "above the linear trend" in early stages of fine-tuning. The OOD accuracy then rises more quickly than the ID accuracy early on, though the final accuracies agree with a linear trend across models.

Most recently, the large-scale study of Wenzel et al. [50] is more nuanced: they observe a linear trend only on some datasets. Their setup consists in fine-tuning an ImageNet-pretrained model on a chosen dataset and evaluating it on ID and OOD splits w.r.t. this dataset. They repeat the procedure with a variety of datasets, architectures, and other implementation choices such as data augmentations. The scatter plots of ID / OOD accuracy in [50] show four typical patterns (Figure 2).

---

[3]The "*linear trend*" is not really linear: it applies to probit-scaled accuracies (a non-linear transform).

[4][22] explains linear trends with (1) data points having similar probabilities of occurring ID and OOD, and (2) a low probability that a model correctly classifies points that a higher-accuracy model misclassifies.

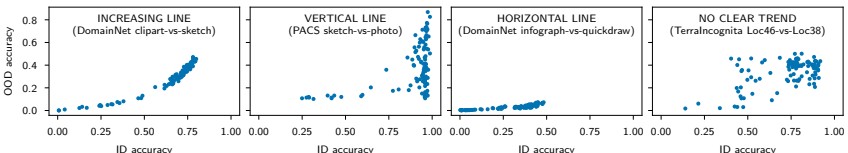

Figure 2: Typical patterns observed in [50] (reproduced with permission).

1. **Increasing line (positive correlation): mild distribution shift.** ID and OOD accuracies are positively correlated. Focusing on classical (ID) generalization brings concurrent OOD improvements.

2. **Vertical line: underspecification** [5, 16, 47]. Different models obtain a similar high ID performance but different OOD performance. The objective of high ID performance does not sufficiently constrains the learning. Typically, multiple features in the data (a.k.a. biased or spurious features) can be used to obtain high ID performance, but not all of them are equally reliable on OOD data. To improve OOD performance, additional task-specific information is necessary, e.g. additional supervision or inductive biases (custom architectures, regularizers, etc.).

3. **Horizontal line, low OOD accuracy: severe distribution shift.** No model performs well OOD. A severe shift prevents any transfer between training and OOD test data. The task needs to be significantly more constrained e.g. with task-specific inductive biases.

4. **No clear trend: underspecification.** Models show a variety of ID and OOD accuracies. The difference with (2) is the wider variety along the ID axis, e.g. because a difficult learning task yields solutions of lower ID accuracy from local minima of the ERM objective.

The authors note the absence of decreasing patterns, which are however possible in theory.

5. **Decreasing line (inverse correlation): misspecification.** The highest accuracy ID and OOD are achieved by different models. Optima of the ERM objective, which are expected to be optima in ID performance, do not correspond to optima in OOD performance. This implies a trade-off: higher OOD performance is possible at the cost of lower ID performance.

---

**When does an inverse correlation occur between ID and OOD performance?**

Intuitively, it can occur when a pattern in the data is predictive in one distribution and misleading in the other. For example, object classes $\mathcal{C}_1$ and $\mathcal{C}_2$ are respectively associated with image backgrounds $\mathcal{B}_1$ and $\mathcal{B}_2$ in ID data, and respectively $\mathcal{B}_2$ in $\mathcal{B}_1$ (swapped) in OOD data. Relying on the background can improve performance on either distribution but not both simultaneously. While such severe shifts might be rare, the next section presents an actual example.

---

## 3 New observations: inversely-correlated ID/OOD performance

This section is an in-depth examination using the WILDS-Camelyon17 dataset [15]. We include experiments on other datasets in Section 5.1 and Appendix B. Here, we use Camelyon17 in a manner similar to Wenzel et al. [50]. These authors evaluated different architectures and assumed that their different inductive biases can produce models that cover a range of ID/OOD accuracies. In contrast and for simplicity, we rely instead on different random seeds since [5] showed that this is sufficient to

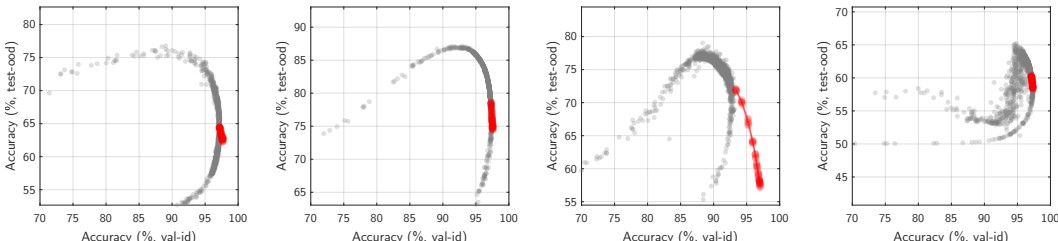

Figure 3: Our new observations show that higher OOD accuracy can sometimes be traded for lower ID accuracy. Each panel corresponds to a different pretraining seed. Each dot represents a linear classifier on frozen features, re-trained with a different seed and/or number of epochs. They are re-trained with standard ERM (red dots ●) or a diversity-inducing method (gray dots ●). The latter set includes models with higher OOD/lower ID accuracies. See Appendix A for additional plots.

cover a variety of ID/OOD accuracies on this dataset. We also want to minimize the experimenter's bias. Therefore, to further increase variety without the manual arbitrary selection of architectures of [50], we also train models with the general-purpose, diversity-inducing method of [46].

---

**Background: learning diverse solutions.**

A range of methods have been proposed to train multiple networks to similar ID performance while differing in other properties such as OOD generalization. These "diversification" methods are relevant in cases of underspecification [5] when the standard ERM objective does not constrain the solution space to a unique one. Recent methods train multiple models (in parallel or sequentially) while encouraging diversity in **feature space** [11, 57], **prediction space** [28, 16], or **gradient space** [36, 35, 46, 47]. **We use method of [46] that encourages gradient diversity**.

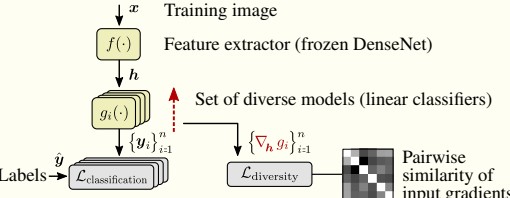

Figure 4: Method used to train a diverse set of models. Each training image $x$ goes through a frozen pretrained DenseNet to produce features $h=f(x)$. We train a set of linear classifiers $\{g_i\}_{i=1}^n$ on these features. A diversity loss minimizes the pairwise similarity between their input gradients.

The method trains many copies of the same model in parallel – in our case, a linear classifier on top of a frozen DenseNet backbone (see Figure 4). The models are optimized by standard SGD to minimize the sum of a standard classification loss (cross-entropy) with a diversity loss that encourages diversity across models. Using $\lambda$ a weight hyperparameter, the complete loss is $\mathcal{L}=\mathcal{L}_{\text{classification}} + \lambda \mathcal{L}_{\text{diversity}}$. The second term encourages each copy to rely on different features by minimizing the mutual alignment of input gradients:

$$\mathcal{L}_{\text{diversity}} = \Sigma_{x \in \text{Tr}} \, \Sigma_{i=1}^n \, \Sigma_{j=i+1}^n \, \nabla_h \, g_i(h).\nabla_h \, g_j(h), \quad \text{with } h = f(x). \tag{1}$$

These pairwise dot products quantify the mutual alignment of the gradients. Intuitively, minimizing (1) makes each model locally sensitive along different directions in its input space.

Assuming that $g$ produces a vector of logits (as many as there are classes), $\nabla_h \, g(\cdot)$ refers to the gradient of the largest logit w.r.t. the classifier's input $h$. We use $n=24$ copies and a weight $\lambda=10$ that were selected for giving a wide range of ID accuracies. See [46] for details about the method.

---

**Experimental details.** We use 10 DenseNet-121 models pretrained by the authors of the dataset with different seeds [15]. For each, we re-train the last linear layer from a random initialization for 1 to 10 epochs, keeping other layers frozen. These are referred to as **ERM models**. We perform this re-training with 10 different seeds which gives $10^3$ ERM models (10 pretraining seeds × 10 re-training seeds × 10 numbers of epochs). In addition, we repeat this re-training of the last layer with the diversity-inducing method of [46] (details in the box below). These are referred to as **diverse models**. Each run of the methods produces 24 different models, giving a total of $10^3 \cdot 24$ such models.

**Results with ERM models.** In Figure 3 we plot the ID vs. OOD accuracy of ERM models as red dots (●). Each panel corresponds to a different pretraining seed. The variation across panels (note the different Y-axis limits) shows that OOD performance varies across pre-training seeds even though the ID accuracy is similar, as noted by [15]. Our new observations are visible *within* each panel. The dots (models) in any panel differ in their re-training seed and/or number of epochs. The seeds induce little variation, but the number of epochs produce patterns of decreasing trend (negative correlation). Despite the narrow ID variation (X axis), careful inspection confirms that the pattern appears in nearly all cases (see Appendix A for zoomed-in plots).

**Results with diverse models.** We plot models trained with the diversity method [46] as gray dots (●). These models cover a wider range of accuracies and form patterns that extend those of ERM models. The decreasing trend is now obvious. This trend is clearly juxtaposed with a *rising* trend of positive ID/OOD correlation. This suggests a point of highest OOD performance after which the model overfits to ID data. Appendix A shows similar results with other pretraining seeds. The patterns are

not always clearly discernible because large regions of the performance landscape are not covered, despite the diversity-inducing method. We further discuss this issue next.

## 4 Past studies missed negative correlations due to biased sampling of models

We identified several factors explaining the discrepancy between our observations and past studies.

- ERM models alone do not always form clear patterns (red dots ● in Figure 3). In our observations, the **models trained with a diversity-inducing method** (gray dots ●) were key in making the suspected patterns more obvious, because they cover a wider range of accuracies.

- The ID/OOD trade-off varies during training, as noted by [2]. This **variation across training epochs** is responsible for much of the newly observed patterns. However, models of different architectures or pretraining seeds are not always comparable with one another because of shifts in their overall performance (see e.g. different Y-axis limits across panels in Figure 3). Therefore the performance across epochs should be analyzed individually per model.

- The "inverse correlation" patterns are not equally visible with all **pretraining seeds**. In some cases, a careful examination of zoomed-in plots is necessary, see Appendix A. This is a reminder that stochastic factors in deep learning can have large effects and that empirical studies should randomize them as much as possible.

To demonstrate these points, we plot our data (same as in Figure 3) while keeping only the ERM models trained for 10 epochs and including all pretraining seeds on the same panel. Figure 5 shows that these small changes reproduce the vertical line observed by Wenzel et al. [50], which completely misses the inverse correlations patterns visible in Figure 3.

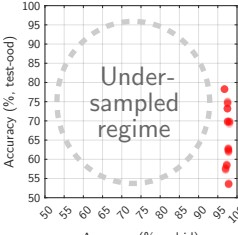

Figure 5: We plot again the ERM models of Figure 3 (red dots ●) but **only include models trained for a fixed number of epochs** and combine all pretraining seeds in the same plot. This reproduces the vertical line from [50], which completely misses the patterns of inverse correlation.

A general explanation is that past studies **undersample regions of the ID/OOD performance space**. They usually consider a variety of architectures in an attempt to sample this space. However, different architectures do not necessarily behave very differently from one another (see the box below). We lack methods to reliably identify models of high OOD performance, but the diversity-inducing method that we use yields models spanning a wide range of the performance spectrum.

> **Why isn't it sufficient to evaluate a variety of architectures?**
> Different architectures do not necessarily induce radically different behaviour [39]. Even CNNs and vision transformers have similar failure modes [29]. Distinct architectures can share similar inductive biases due e.g. to SGD, the simplicity bias [39, 40], or neural anisotropies [27].

## 5 Occurrences in other datasets

In addition to Camelyon17, we performed experiments on five additional datasets used in OOD research and found inverse correlations on four out of five. We present two of them below and the others in Appendix B.

In these experiments, we train standard architectures with well-known methods: standard ERM, simple class balancing [12], mixup [60], selective mixup [56], and post hoc adjustment for label shift [19] (we did not use the diversification method from Section 3). We repeat every experiment with 10 seeds and plot the ID/OOD accuracy from every epoch in the following figures.

## 5.1 WildTime-arXiv

**Data.** The WildTime-arXiv [55] dataset contains text abstracts from arXiv preprints. The task is to predict each paper's category among 172 classes. The ID and OOD splits are made of data from different time periods.

**Methods.** We fine-tune a standard BERT-tiny model with a new linear head, using any of these well-known methods: standard ERM, simple class balancing [12], mixup [60], selective mixup [56], and post hoc adjustment for label shift [19] (we did not use the diversification method from Section 3). We repeat every experiment with 10 seeds and record the ID and OOD accuracy at every training epoch. We then plot each of these points in Figure 6 and highlight the epoch of highest ID or OOD accuracy per run (method/seed combination).

**Results.** As discussed in Section 5.1, there is a clear trade-off both within methods (i.e. across seeds and epochs) and across methods.

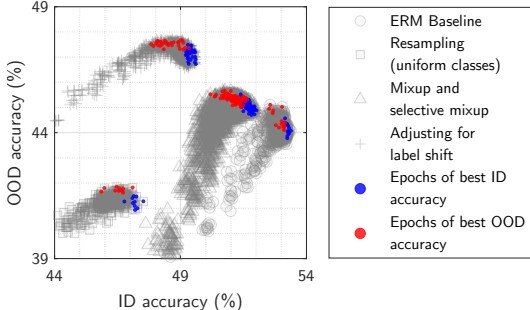

Figure 6: Results on WildTime-arXiv.

## 5.2 Waterbirds

**Data.** The **waterbirds** dataset [37] is a synthetic dataset widely used for evaluating OOD generalization. The task is to classify images of birds into 2 classes. The image backgrounds are also of two types, and the correlation between birds and background is reversed across the training and test splits. The standard metric is the worst-group accuracy, where each group is any of the 4 combinations of bird/background.

**Methods.** We follow the same procedure as described above. We experiment two classes of architectures: ResNet-50 models pretrained on ImageNet and fine-tuned on waterbirds, and linear classifiers trained of features from the same frozen (non-fine-tuned) ResNet-50.

**Results.** We observe in Figure 7 patterns of inverse correlations in both cases.

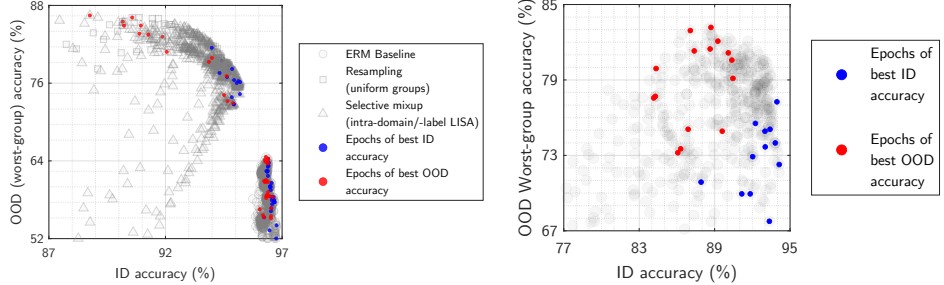

Figure 7: Results on waterbirds with linear probing (left) and fine-tuned ResNet-50 models.

## 5.3 Occurrences in the existing literature

A literature review reveals other cases across a range of topics, many of which were not particularly highlighted by their authors and required close examination.

- A close examination of [9, Table 2] reveals a clear inverse pattern on three benchmarks for natural language inference (NLI). This task is known for biases and shortcuts in the training data, and the OOD test sets in these benchmarks correspond to severe distribution shifts. Our

proposed explanation (right end of the spectrum in Figure 8) therefore aligns with these observations. Experiments on question answering from the same authors use data with milder distribution shifts. Correspondingly, they show instead a positive correlation.

- Kaplun et al. [14, Figure 7] find that the CIFAR dataset contains a subset (CIFAR-10-Neg) on which the performance of visual classifiers is inversely correlated with their ID performance.
- Xie et al. [53, Section 5.3] discuss cases of inverse correlation on the CelebA dataset with their In-N-Out method – albeit within an overall positive trend.
- McCoy et al. [23] show that BERT models trained with different seeds vary widely in performance on the HANS benchmark for NLI while their ID performance on MNLI is similar.
- Naganuma et al. [26] performed an extensive evaluation on OOD benchmarks after the initial release of this paper. They consider a wider range of hyperparameters than existing works, and as expected from our claims, they observe a broader range of ID/OOD relations than the "linear trend".
- Liang et al. [18] focused on datasets with subpopulation shifts and analyzed the relation between performance across subgroups. They also find a linear trend to be an inaccurate description. They observe non-linear relations with a transition point between models showing a negative correlation and others showing a positive one.
- In their paper on *Model recycling*, Ramé et al. [31] include plots of ID/OOD performance (Appendix A) that are nothing like linear correlations. Instead, bell-shaped curves unambiguously indicate a necessary trade-off between ID and OOD performance.
- Work on adversarial examples has examined the trade-off between standard and adversarially-robust accuracy [30, 54, 59]. This agrees with our explanations (Figure 8) since adversarial inputs correspond to extreme distribution shifts.
- The literature on transfer learning has previously shown occasional cases of negative transfer, a related phenomenon where improving performance in one domain hurts in others [20, 49].

## 6  Theoretical analysis

The theoretical possibility of an inverse ID/OOD correlation is quite obvious. In Appendix C), we show below theoretically how to construct a toy example where ID and OOD are negatively correlated. We then explain how ID and OOD metrics can diverge as training progresses, or as one trains a model on data containing more and more spurious features.

However, the theoretical analysis is not central to our argument. The more important open question is whether these cases happen merely in pathological situations, or also in real-world data. Prior work strongly argued for the former, but we show that the latter is actually correct. The key contribution of this paper is therefore in the observations on actual data, rather than in a theoretical argument.

## 7  Ordering ID/OOD patterns according to shift magnitude

The above analysis shows that inverse correlation patterns are essentially due to the presence of spurious features, i.e. features whose predictive relation with the target in ID data becomes misleading OOD. Occurrences of spurious features increase with the magnitude of the distribution shift. Therefore, the possible patterns in ID/OOD performance presented in Section 2 can be ordered according to the magnitude of the distribution shift they are likely to occur with (see Figure 8).

With the smallest distribution shifts (leftmost case in Figure 8), for example training on ImageNet and testing on its replication ImageNet v2 [32], ID validation performance closely correlates with OOD test performance. This OOD setting is the easiest because one can focus on improving classical generalization and reap concurrent improvements OOD.

With a larger distribution shift, more features are likely to be spurious, which is likely to break the ID/OOD correlation. The task of improving OOD performance is likely to be under- or misspecified, i.e. there is not enough information to determine which features a model should rely on.

Valid approaches include modifying the training objective, injecting task-specific information (e.g. building-in invariance to rotations as in [43]), well-chosen data augmentations, or inhomogeneous training data such as multiple training environments [17] or counterfactuals [44, 1].

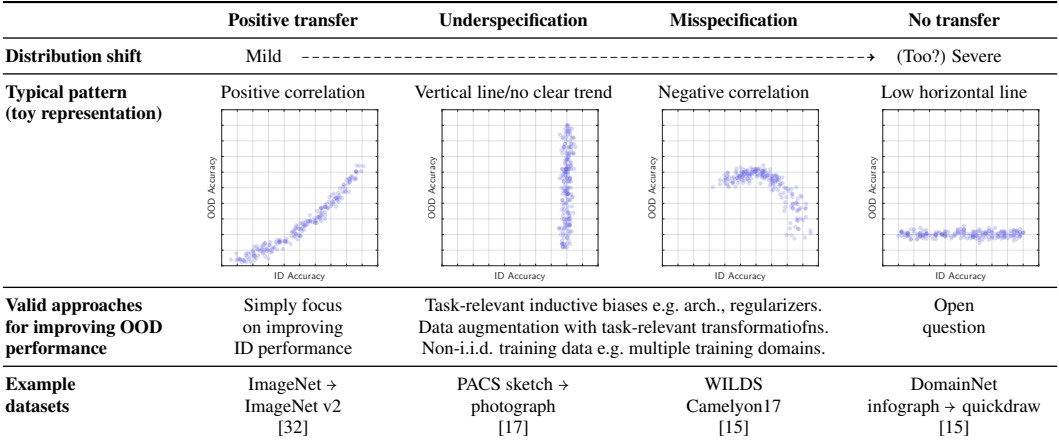

| | Positive transfer | Underspecification | Misspecification | No transfer |
|---|---|---|---|---|
| **Distribution shift** | Mild | -------------------------------------------------------→ | | (Too?) Severe |
| **Typical pattern (toy representation)** | Positive correlation | Vertical line/no clear trend | Negative correlation | Low horizontal line |
| **Valid approaches for improving OOD performance** | Simply focus on improving ID performance | Task-relevant inductive biases e.g. arch., regularizers. Data augmentation with task-relevant transformatiofns. Non-i.i.d. training data e.g. multiple training domains. | | Open question |
| **Example datasets** | ImageNet → ImageNet v2 [32] | PACS sketch → photograph [17] | WILDS Camelyon17 [15] | DomainNet infograph → quickdraw [15] |

Figure 8: Overview of ID vs. OOD patterns occurring at different levels of distribution shift.

With extreme distribution shifts, most predictive features are overwhelmingly spurious and it is very difficult to learn any one relevant in OOD data (rightmost case in Figure 8).

The proposed ordering of patterns is rather informal and could be further developed following the two axes of diversity shifts and correlation shifts proposed by [58] (see also [52]). More recently, [48] showed that the suitability of various methods for OOD generalization depends on particularities of the underlying causal structure of the task – which must therefore be known to select a suitable method. Identifying which ID / OOD patterns occur with particular causal structures might serve as a tool to understand the type of OOD situation one is facing and identify a suitable method.

## 8 Revisiting advice from past studies

We have established that observations in past studies were incomplete. We now bring nuance to some recommendations and conclusions made in these studies.

- **Focusing on a single metric.**

  > "*We see the following potential prescriptive outcomes (...) correlation between OOD and ID performance can simplify model development since **we can focus on a single metric**.*" [25]

  We demonstrated that inverse correlations do occur, hence there exist scenarios where ID performance is misleading. Relying on a single metric for model development is ill-advised [45] especially if it cannot capture necessary trade-offs. We recommend tracking multiple metrics e.g. performance on multiple distributions or interpretable predictions on representative test points.

- **Improving ID performance for OOD robustness.**

  > "*If practitioners want to make the model more robust on OOD data, **the main focus should be to improve the ID classification error**. (...) We speculate that the risk of overfitting large pretrained models to the downstream test set is minimal, and it seems to be not a good strategy to, e.g., reduce the capacity of the model in the hope of better OOD generalization.*" [50]

  This recommendation assumes the persistence of a positive correlation. On the opposite, we saw that a positive correlation can precede a regime of inverse correlation (Figure 3, left panels). If the goal is to improve OOD, focusing on ID performance is a blind alley since this goal requires to increase ID performance at times and reduce it at others.

- **Future achievable OOD performance.**
  As obvious as it is, it feels necessary to point out that empirical studies only chart regimes achievable with existing methods. Observations have limited predictive power, hence more care seems warranted when deriving prescriptive recommendations from empirical evidence.

  The best possible performance e.g. on Camelyon17 is obviously not limited to the Pareto front of our experiments. The state of the art on this dataset [33, 51] injects additional task knowledge to bypass the under/misspecification of ERM, and exceeds both our highest ID and OOD performance.

The important message remains that a given hypothesis class (DenseNet architecture in our case) admits parametrizations on which ID and OOD metrics do not necessarily correlate.

- **Possible invalidation of existing studies.**

  The possibility of inverse correlations may invalidate studies that implicitly assume a positive one. For example, Angarano et al. [3] evaluate the OOD robustness of computer vision backbones. They find modern architectures surpass domain generalization methods. However, they discard any model with submaximal ID performance by performing "*training domain validation*" as in [10]. Any model with high OOD performance but suboptimal ID is ignored. They also train every model for a fixed, large number of epochs. And this may additionally prevent from finding models with high OOD performance since robustness is often progressively lost during fine-tuning [2].

  By design, this study [3] is incapable of finding OOD benefits of architectures or methods that require trading off some ID performance. Most importantly, once the assumption of a positive correlation is enacted by throwing away models with submaximal ID performance, there is no more opportunity to demonstrate its validity.

  An even more recent example of this fallacy is found in [21]. The authors construct a new OOD benchmark, train and tune baselines for maximum ID performance, then observe:

  > "*We find no [positive nor inverse] correlation between in- and out-of-distribution environment performance. All methods consistently achieve 98-99% ID test performance.*" [21]

  But the authors did not give the chance to make any other observation. As in Figure 5, the ID criterion means that we only get to observe a thin vertical slice of the ID vs. OOD plot.[5]

## 9 Discussion

This paper highlighted that inverse correlations between ID / OOD performance are possible, not only theoretically but also with real-world data. It is difficult to estimate how frequent this situation is. Although we examined a single counterexample, we also showed that past studies may have systematically overlooked such cases. This suffices to show that one cannot know a priori where a task falls on the spectrum of Figure 8. It is clearly ill-advised to blindly assume a positive correlation.

**Can we avoid inverse correlations with more training data?** Scaling alone without data curation seems unlikely to prevent inverse correlations. [8] examined a more general question and determined that the impressive robustness of the large vision-and-language model CLIP is determined by the *distribution* of its training data rather than its quantity. Similarly, inverse correlations stem from biases in the training distribution (e.g. a class $\mathcal{C}_1$ appearing more frequently with image background $\mathcal{B}_1$ than any other). And biases in a distribution do not vanish with more i.i.d. samples. Indeed, more data can cover more of the support of the distribution, but this coverage will remain uneven, i.e. biased. The problem can become one of "subpopulation shift" [38] rather than distribution shift, but it remains similarly challenging.

**Training full networks with a diversity-inducing method.** We showed inverse correlations with standard ERM models and with linear classifiers trained with a diversity-inducing method [46]. To the best of our knowledge, this diversity method has not been applied to deep models because of its computational expense. It would be interesting to confirm our observations on networks trained entirely with diversity-inducing methods.

**Qualitative differences along the Pareto frontier.** Besides quantitative performance, interpretability methods could examine whether various ID / OOD trade-off models rely on different features and generalization strategies as done in NLP in [13].

**Model selection for OOD generalization** has recently seen promising advances. [7] get around selection based on either ID or OOD validation data with a tunable trade-off in their Quantile Risk Minimization method. And [48] examined existing approaches to OOD generalization from their suitability to various distribution shifts and causal structures.

---

[5]Statements in [21] are factually correct but misleading. They are equivalent to "*we did not detect X*" but leave it to the reader to figure out "*we designed our experiments such that there is no way of detecting X*".

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

# Appendix

## A Additional results on Camelyon17

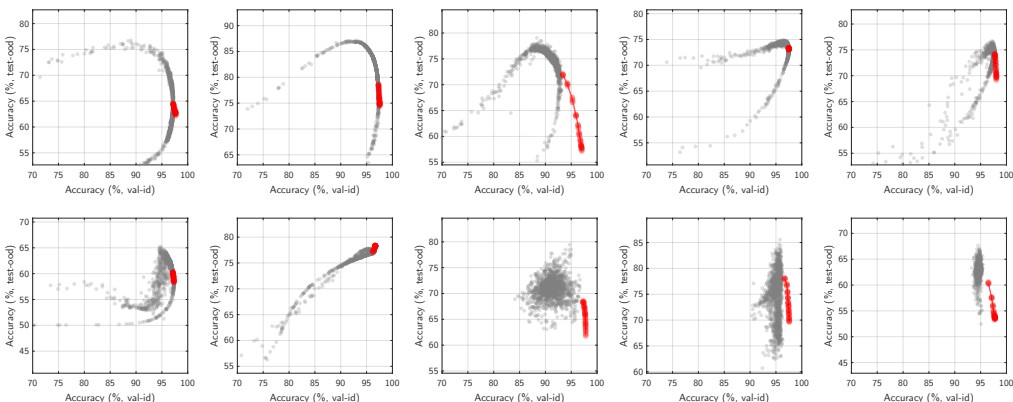

Figure 9: As in Figure 3, we show that higher OOD accuracy can be sometimes be traded off for a lower ID accuracy. Each panel shows results from a different pretrained model (i.e. pretrained with a different random seed). Each dot represents a linear classifier re-trained on features from this pretrained model with standard ERM (red dots •) or with a diversity-inducing method [46] (gray dots •). The latter set includes models with higher OOD / lower ID accuracies.

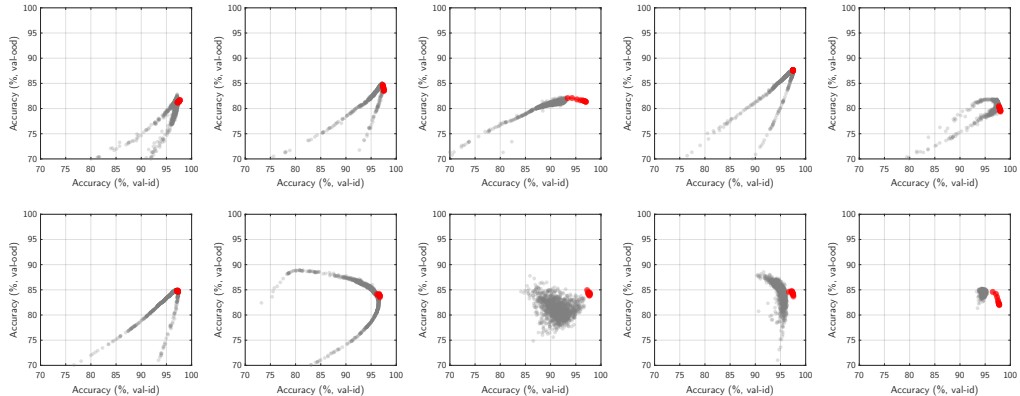

Figure 10: Same as in Figure 9, using `val-ood` instead of `test-ood` as the OOD evaluation set.

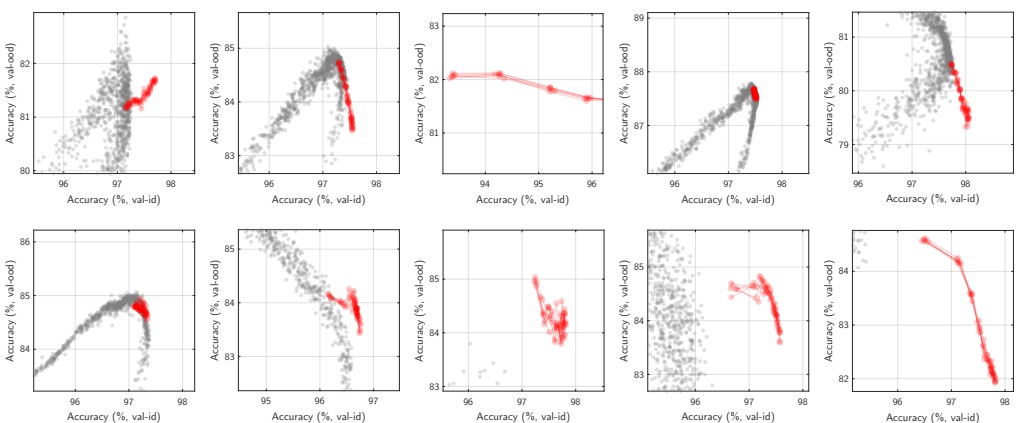

Figure 11: Same as in Figure 10, zoomed-in on ERM models (red dots ●).

## B    Results on other datasets

### B.1    CivilComments

**Data.** The CivilComments dataset [15] is another widely-used dataset in OOD research. It contains text comments from internet forums to be classified as toxic or not. Each example is labeled with a topical attribute (e.g. Christian, male, LGBT, etc.) that is spuriously associated with ground truth labels in the training data. The target metric is again worst-group accuracy, where a group is any label/attribute combination.

**Methods.** We follow the same procedure as described above. We experiment two classes of architectures: pretrained BERT-tiny fine-tuned on CivilComments, and linear classifiers trained of features from the same frozen BERT-tiny models (a.k.a. linear probing).

**Results.** We observe in Figures 12–13 different patterns with the two classes of architectures. With linear probing, the ID vs. OOD trade-off is minimal, and the model of highest ID performance within a run as well as across methods is very similar to the model of highest OOD performance. With fine-tuning however, the trade-off is more pronounced. The ID and OOD performance usually peak then diminish at different epochs during the fine-tuning. This agrees with previous reports [2] of OOD robustness being progressively lost during fine-tuning.

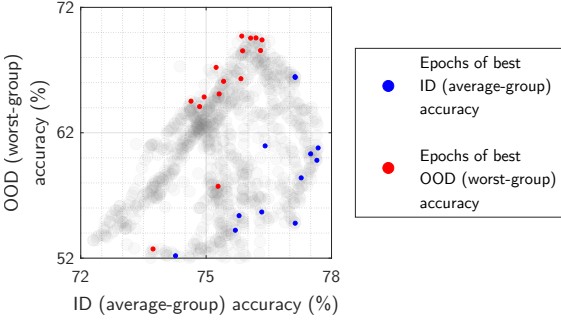

Figure 12: Results on CivilComments with fine-tuned BERT.

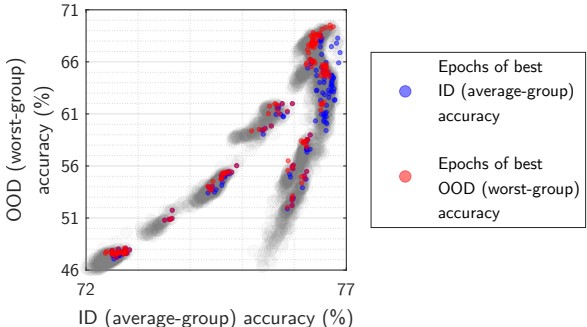

Figure 13: Results on CivilComments with linear probing on frozen BERT embeddings.

### B.2    WildTime-MIMIC-Readmission

**Data.** The WildTime-MIMIC-Readmission [55] dataset contains hospital records (sequences of codes representing diagnoses and treatments) to be classified into two classes, corresponding to the readmission of the patient within a short time. ID and OOD splits contain records from different time periods.

**Methods.** We follow the same procedure as described above. We train a standard bag-of-embeddings architecture, which associate each diagnosis/treatment with a learned embedding, then summed and fed to a linear classifier. We train this model with standard ERM, and with a resampling to balance the classes in the training data [12], which is a standard approach for imbalanced datasets. We also

train models with a "mild balancing", where classes are sampled according to a distribution half-way between the original one of the training data, and a uniform (50%–50%) one.

**Results.** In Figure 14 we observe that ID and OOD performance are mostly positively correlated across methods. The best models are obtained with uniform balancing of classes, in which case model selection based on OOD performance could give a small advantage, but it is marginal compared to the improvement over the ERM baseline, which can be clearly detected on both the ID and OOD performance.

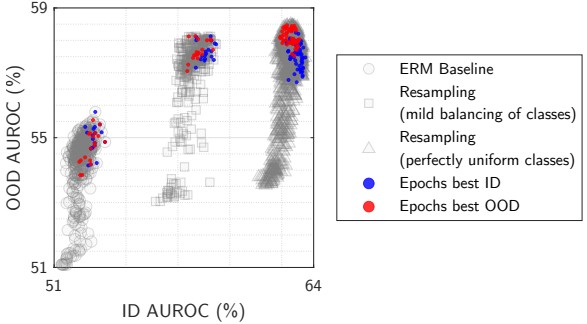

Figure 14: Results on WildTime-MIMIC-Readmission.

### B.3 WildTime-Yearbook

**Data.** The WildTime-Yearbook [55] dataset contains yearbook portraits. Each image is to be classified as male or female, and the ID and OOD splits contain images from different time periods.

**Methods.** We follow the same procedure as described above. We train the simple CNN architecture described in [55]. We report in Figure 15 both the "average-group" accuracy (over the entire OOD test set) and the "worst-group" accuracy (where a group is any 5-year period within the OOD test period.

**Results.** The patterns are slightly different in the two cases but similar conclusions can be drawn from both. There is a mostly-positive correlation, but at the highest accuracies (upper-right quadrant), some small trade-off exists. This suggests that fine-grained differences exist that are useful for either ID or OOD generalization, but not both. Although differences are small, this "pointy end" of the spectrum is where the state-of-the-art models compete, hence the relevance of this observation.

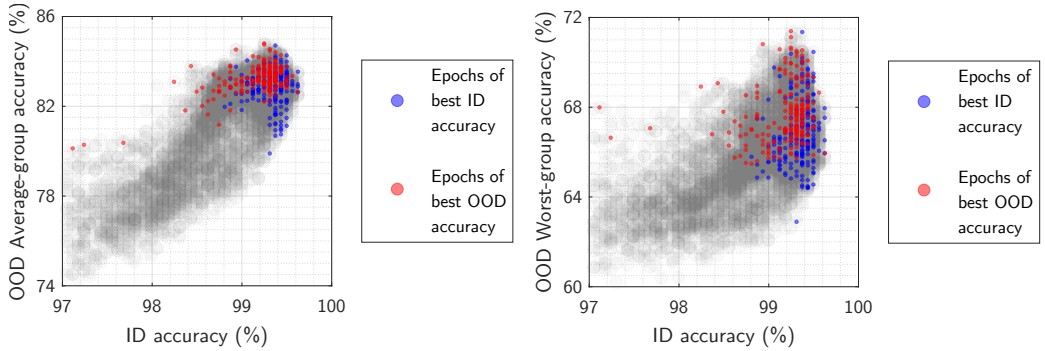

Figure 15: Results on WildTime-Yearbook (left: average-group accuracy, right: worst-group accuracy).

## C  Theoretical analysis

We present a minimal case of a trade-off between ID and OOD performance to aid understanding its cause and show it can occur even in a simple linear setting. Let $y \in \mathbb{R}$ be the target variable predicted by the model and $x$ the features used as input. Without loss of generality, we consider a regression setting for mathematical convenience, though the results are valid for classification as well. The purpose of this analysis is to investigate theoretically how variations in certain input features correspond to changes in risk (i.e. expected loss) on OOD data. We consider the input features to be a concatenation of invariant and spurious elements: $x = [x_{\mathrm{inv}} ; x_{\mathrm{spu}}]$ with $x_{\mathrm{inv}} \in \mathbb{R}^{d_{\mathrm{inv}}}$ and $x_{\mathrm{spu}} \in \mathbb{R}^{d_{\mathrm{spu}}}$. Following [4, 34, 61], we consider the simple data-generating process defined by the following structural equations: $y^e := \gamma^\top x_{\mathrm{inv}}^e + \epsilon_{\mathrm{inv}}$, and $x_{\mathrm{spu}}^e := y^e \mathbf{1}^s + \alpha^e \circ \epsilon_{\mathrm{spu}}$ where $e \in \{e_{\mathrm{ID}}, e_{\mathrm{OOD}}\}$ is an environment index referring to ID or OOD data, $\mathbf{1}^s$ is a vector of ones and $\circ$ denotes the element-wise product. The random variables $\epsilon_{\mathrm{inv}}$ and $\epsilon_{\mathrm{spu}}$ represent symmetric independent random noise with zero-mean, sub-Gaussian tail probabilities such that $\mathrm{Var}(\epsilon_{\mathrm{inv}}) > 0$, $\mathrm{Var}(\epsilon_{\mathrm{spu},i}) > 0$, $\forall i \in [1, d_{\mathrm{spu}}]$. The vector $\gamma \in \mathbb{R}^{d_{\mathrm{inv}}}$ determines the linear relation between the target variable and the invariant features, which is identical across environments. In contrast, the vector $\alpha^e$ acts on the spurious features and is environment-specific. Invariant features are similarly predictive on ID and OOD data while spurious ones are not.

Let us now study the relationship between ID and OOD performance of a predictive model that relies on a subset $\Phi$ of the features $x$ (not that the subset could be spurious or invariant). This subset is represented with a binary mask $\Phi \in \{0,1\}^{d_{\mathrm{inv}} + d_{\mathrm{spu}}}$. Suppose we have selected $\hat{d}_{\mathrm{inv}}$ invariant features and $\hat{d}_{\mathrm{spu}}$ spurious features, with $(\hat{d}_{\mathrm{inv}} + \hat{d}_{\mathrm{spu}}) = \hat{d} = ||\Phi_{\hat{d}}||_1$ the number of selected features. We denote the selected features by $\Phi_{\hat{d}}$ with $[x_{\mathrm{inv},1}, ..., x_{\mathrm{inv},\hat{d}_{\mathrm{inv}}}, x_{\mathrm{spu},1}, ..., x_{\mathrm{spu},\hat{d}_{\mathrm{spu}}}]$. To investigate the changes in model performance, we consider a simple linear regression model. This enables us to theoretically measure the sensitivity of predictions when additional spurious features are added without complications from non-linearities. We denote with $\mathbb{E}^{\mathrm{ID}}$ and $\mathbb{E}^{\mathrm{OOD}}$ the in- and out-of-domain expectation, and with $\beta$ the optimal parameters of the linear regression for a certain domain, i.e. $\beta_{\hat{d}} = \mathbb{E}[\Phi_{\hat{d}}(x)^\top \Phi_{\hat{d}}(x)] \, \mathbb{E}[\Phi_{\hat{d}}(x)^\top y]$. The MSE loss of the fitted linear regressor is then $\mathbb{E}[y - \Phi_{\hat{d}}(x)^\top \beta_{\hat{d}}]^2$. Given the feature mask $\Phi_{\hat{d}}$, we denote with $\mathcal{L}_{\mathrm{ID}}(\Phi_{\hat{d}})$ and $\mathcal{L}_{\mathrm{OOD}}(\Phi_{\hat{d}})$ the risk of the model on the in and out-of-domain distribution, respectively. Theorem 1 below (see complete form including assumptions and proof in Appendix D) examines how the ID and OOD risks of the model vary as additional spurious features are included into $\Phi_{\hat{d}}$ to obtain $\Phi_{\hat{d}+1}$.

**Theorem 1.** *Under Assumption 1 and sufficient changes in $\alpha$, including an additional spurious feature leads to:*

$$\mathcal{L}_{\mathrm{ID}}(\Phi_{\hat{d}+1}) - \mathcal{L}_{\mathrm{ID}}(\Phi_{\hat{d}}) = \mathbb{E}^{\mathrm{ID}}[y - \Phi_{\hat{d}}(x)^\top \beta_{\hat{d}}^{\mathrm{ID}}]^2 - \mathbb{E}^{\mathrm{ID}}[y - \Phi_{\hat{d}+1}(x)^\top \beta_{\hat{d}+1}^{\mathrm{ID}}]^2 < 0$$
$$\mathcal{L}_{\mathrm{OOD}}(\Phi_{\hat{d}+1}) - \mathcal{L}_{\mathrm{OOD}}(\Phi_{\hat{d}}) = \mathbb{E}^{\mathrm{OOD}}[y - \Phi_{\hat{d}}(x)^\top \beta_{\hat{d}}^{\mathrm{OOD}}]^2 - \mathbb{E}^{\mathrm{OOD}}[y - \Phi_{\hat{d}+1}(x)^\top \beta_{\hat{d}+1}^{\mathrm{OOD}}]^2 > 0 \,.$$

Theorem 1 shows that adding a spurious feature to those used by the model can affect its ID and OOD losses in opposite directions, implying a trade-off between ID and OOD accuracy. In other words, this minimal case shows that a simple model without/with an extra (spurious) feature can exhibit an inverse correlation between its ID and OOD performance.

## D  Proof of Theorem 1

**Theorem 1.** Including an additional spurious feature leads to the following change in the risks:

$$
\begin{aligned}
\mathcal{L}_{\mathrm{ID}}(\Phi_{\hat{d}+1}) \quad &- \quad \mathcal{L}_{\mathrm{ID}}(\Phi_{\hat{d}}) \quad = \quad \mathbb{E}^{\mathrm{ID}}[y - \Phi_{\hat{d}}(x)^\top \beta_{\hat{d}}^{\mathrm{ID}}]^2 \quad - \quad \mathbb{E}^{\mathrm{ID}}[y - \Phi_{\hat{d}+1}(x)^\top \beta_{\hat{d}+1}^{\mathrm{ID}}]^2 \quad < \quad 0 \\
\mathcal{L}_{\mathrm{OOD}}(\Phi_{\hat{d}+1}) \quad &- \quad \mathcal{L}_{\mathrm{OOD}}(\Phi_{\hat{d}}) \quad = \quad \mathbb{E}^{\mathrm{OOD}}[y - \Phi_{\hat{d}}(x)^\top \beta_{\hat{d}}^{\mathrm{OOD}}]^2 \quad - \quad \mathbb{E}^{\mathrm{OOD}}[y - \Phi_{\hat{d}+1}(x)^\top \beta_{\hat{d}+1}^{\mathrm{OOD}}]^2 \quad > \quad 0 \\
\mathcal{L}_{\mathrm{OOD}}(\Phi_{\hat{d}+1}) \quad &- \quad \mathcal{L}_{\mathrm{OOD}}(\Phi_{\hat{d}}) \quad = \quad Q_1 + Q_2 + Q_3
\end{aligned}
$$

with $Q_1$, $Q_2$, $Q_3$ defined as:

$$Q_1 = \mathbb{E}^{\text{OOD}}[y - \Phi_{\hat{d}}(\boldsymbol{x})^\top \beta_{\hat{d}}^{\text{OOD}}]^2 - \mathbb{E}[y - \Phi_{\hat{d}+1}(\boldsymbol{x})^\top \beta_{\hat{d}+1}^{\text{OOD}}]^2$$

$$Q_2 = \sum_{i=1}^{\hat{d}} \left[ \left( \mathbb{E}^{\text{OOD}}[\Phi_{\hat{d}}(\boldsymbol{x})y]^\top \boldsymbol{v}_i^{\text{OOD},\hat{d}} \right)^2 \left( \lambda_i^{\text{OOD},\hat{d}} \right) \left( \frac{1}{\lambda_i^{\text{ID},\hat{d}}} - \frac{1}{\lambda_i^{\text{OOD},\hat{d}}} \right)^2 \right.$$

$$\left. - \left( \mathbb{E}^{\text{OOD}}[\Phi_{\hat{d}}(\boldsymbol{x})y]^\top \boldsymbol{v}_i^{\text{OOD},\hat{d}+1} \right)^2 \left( \lambda_i^{\text{OOD},\hat{d}+1} \right) \left( \frac{1}{\lambda_i^{\text{ID},\hat{d}+1}} - \frac{1}{\lambda_i^{\text{OOD},\hat{d}+1}} \right)^2 \right]$$

$$Q_3 = \left( \mathbb{E}^{\text{OOD}}[\Phi_{\hat{d}+1}(\boldsymbol{x})y]^\top \boldsymbol{v}_{\hat{d}+1}^{\text{OOD},\hat{d}+1} \right)^2 \frac{((\alpha_{\hat{d}+1}^{\text{ID}})^2 - (\alpha_{\hat{d}+1}^{\text{OOD}})^2)^2}{(\lambda_{\hat{d}+1}^{\text{ID},\hat{d}+1})^2 \, \lambda_{\hat{d}+1}^{\text{OOD},\hat{d}+1}} \quad > \quad 0.$$

Further, if the new feature is sufficiently unstable in the test domain, i.e. if $((\alpha_{\hat{d}+1}^{\text{ID}})^2 - (\alpha_{\hat{d}+1}^{\text{OOD}})^2)^2$ is sufficiently large such that:

$$|(\alpha_{\hat{d}+1}^{\text{ID}})^2 - (\alpha_{\hat{d}+1}^{\text{OOD}})^2| \quad > \quad \sqrt{\frac{(\lambda_{\hat{d}+1}^{\text{ID},\hat{d}+1})^2 \lambda_{\hat{d}+1}^{\text{OOD},\hat{d}+1}}{\left( \mathbb{E}^{\text{OOD}}[\Phi(\boldsymbol{x})y]^\top \boldsymbol{v}_{\hat{d}+1}^{\text{OOD},\hat{d}+1} \right)^2}} \ |Q_1 + Q_2|,$$

then we have $Q_3 > |Q_1 + Q_2|$ and therefore $\mathcal{L}_{\text{OOD}}(\Phi_{\hat{d}+1}) - \mathcal{L}_{\text{OOD}}(\Phi_{\hat{d}}) > 0$.

Let $\boldsymbol{x}_{\hat{d}} := \Phi_{\hat{d}}(\boldsymbol{m}x)[\boldsymbol{x}_{\text{inv},1}, ..., \boldsymbol{x}_{\text{inv},\hat{d}_{\text{inv}}}, \boldsymbol{x}_{\text{spu},1}, ..., \boldsymbol{x}_{\text{spu},\hat{d}_{\text{spu}}}]$ be the $\hat{d}$ features already selected, $\boldsymbol{x}_{\hat{d}+1} := \Phi_{\hat{d}+1}(\boldsymbol{m}x)$ the features after adding a new spurious feature $\boldsymbol{x}_{\text{spu},\hat{d}_{\text{spu}}+1}$ to $\boldsymbol{x}_{\hat{d}}$, $[\lambda_1^{\hat{d}}, \lambda_2^{\hat{d}}, ..., \lambda_{\hat{d}}^{\hat{d}}]$ the eigenvalues of $\mathbb{E}[\boldsymbol{x}_{\hat{d}}^\top \boldsymbol{x}_{\hat{d}}]$ and $[\boldsymbol{v}_1^{\hat{d}}, \boldsymbol{v}_2^{\hat{d}}, ..., \boldsymbol{v}_{\hat{d}}^{\hat{d}}]$ the corresponding eigenvectors.

**Assumption 1.** The projection of $\mathbb{E}[\boldsymbol{x}_{\hat{d}}^\intercal \boldsymbol{v}_i^{\hat{d}}]$ on each basis corresponding to feature is non zero, i.e.
$$\left| \mathbb{E}^e[\boldsymbol{x}_{\hat{d}}^\intercal \boldsymbol{v}_i^{\hat{d}}] \right| > 0, \ \ \forall\, e \in \{e_{\text{ID}}, e_{\text{OOD}}\}, \ i \in [d].$$

This ensures that coefficients of a feature can not be always 0, otherwise we can simply remove it.

*Proof.* Let $\beta^{\text{ID}}$ and $\beta^{\text{OOD}}$ denote the solution of linear regression in the ID and OOD domains, i.e.,

$$\beta_{\hat{d}}^{\text{ID}} = \arg \min_\beta \mathbb{E}^{\text{ID}}(y - \boldsymbol{x}_{\hat{d}}^\top \beta)^2 \tag{2}$$

$$\beta_{\hat{d}}^{\text{OOD}} = \arg \min_\beta \mathbb{E}^{\text{OOD}}(y - \boldsymbol{x}_{\hat{d}}^\top \beta)^2 \tag{3}$$

Now let us compare the OOD loss after we include $\boldsymbol{x}_{\text{spu},\hat{d}_{\text{spu}}+1}$. In practice, we can only obtain $\beta^{\text{ID}}$ and then apply it on both the ID and OOD domains, which elicits the following errors:

$$\mathcal{L}_{\text{ID}}(\Phi_{\hat{d}}) = \mathbb{E}^{\text{ID}}(y - \boldsymbol{x}_{\hat{d}}^\top \beta^{\text{ID}}) \tag{4}$$

$$\mathcal{L}_{\text{OOD}}(\Phi_{\hat{d}}) = \mathbb{E}^{\text{OOD}}(y - \boldsymbol{x}_{\hat{d}}^\top \beta^{\text{ID}})$$

$$= \underbrace{\mathbb{E}^{\text{OOD}}(y - \boldsymbol{x}_{\hat{d}}^\top \beta^{\text{ID}}) - \mathbb{E}^{\text{OOD}}(y - \boldsymbol{x}_{\hat{d}}^\top \beta^{\text{OOD}})}_{\xi_1^{\hat{d}}} + \underbrace{\mathbb{E}^{\text{OOD}}(y - \boldsymbol{x}_{\hat{d}}^\top \beta^{\text{OOD}})}_{\xi_2^{\hat{d}}} \tag{5}$$

It is well known that the residual of the linear fitting $y$ by $\boldsymbol{x}_{\hat{d}}$ on the ID domain is

$$\mathcal{L}_{\text{ID}}(\Phi_{\hat{d}}) = \mathbb{E}^{\text{ID}}\left[ y - \boldsymbol{x}_{\hat{d}} \mathbb{E}^{\text{ID}}[\boldsymbol{x}_{\hat{d}}^\top \boldsymbol{x}_{\hat{d}}]^{-1} \mathbb{E}^{\text{ID}}[\boldsymbol{x}_{\hat{d}} y] \right]^2 = \mathbb{E}^{\text{ID}}[y - \Phi_{\hat{d}}(\boldsymbol{x})^\top \beta_{\hat{d}}^{\text{ID}}]^2, \tag{6}$$

Similarly, we have

$$\mathcal{L}_{\text{ID}}(\Phi_{\hat{d}+1}) = \mathbb{E}^{\text{ID}}[y - \boldsymbol{x}_{\hat{d}+1}^\top \beta_{\hat{d}+1}^{\text{ID}}]^2. \tag{7}$$

Since $\boldsymbol{x}_{\text{spu},\hat{d}_{\text{spu}}+1}$ does not lies in the space spaned by $\boldsymbol{x}_{\hat{d}}$, so the space spanned by $\boldsymbol{x}_{\hat{d}+1}$ is strictly larger than $\boldsymbol{x}_{\hat{d}}$.

Together with Assumption 1, we have

$$\mathcal{L}_{\text{ID}}(\Phi_{\hat{d}}) - \mathcal{L}_{\text{ID}}(\Phi_{\hat{d}+1}) = \mathbb{E}^{\text{ID}}[y - \boldsymbol{x}_{\hat{d}}^\top \beta_{\hat{d}}^{\text{ID}}]^2 - \mathbb{E}^{\text{ID}}[y - \boldsymbol{x}_{\hat{d}+1}^\top \beta_{\hat{d}+1}^{\text{ID}}]^2 > 0, \tag{8}$$

and also
$$\xi_2^{\hat{d}} - \xi_2^{\hat{d}+1} = \mathbb{E}^{\mathrm{OOD}}[y - \boldsymbol{x}_{\hat{d}}^\top \beta_{\hat{d}}^{\mathrm{OOD}}]^2 - \mathbb{E}^{\mathrm{OOD}}[y - \boldsymbol{x}_{\hat{d}+1}^\top \beta_{\hat{d}+1}^{\mathrm{OOD}}]^2 > 0. \tag{9}$$

By the proof in Appendix B.6.3 (above Eq. 29) in [61], we have

$$\xi_1^{\hat{d}} = \sum_i^{\hat{d}} (\mathbb{E}^{\mathrm{OOD}}[\boldsymbol{x}_{\hat{d}}^\top y]^\intercal \boldsymbol{v}_i^{\mathrm{OOD},\hat{d}})^2 \lambda_i^{\mathrm{OOD}} (\frac{1}{\lambda_i^{\mathrm{IID},\hat{d}}} - \frac{1}{\lambda_i^{\mathrm{OOD},\hat{d}}})^2. \tag{10}$$

By Eq. (20) in [61], we have

$$\lambda_i^{\mathrm{IID},\hat{d}} - \lambda_i^{\mathrm{OOD},\hat{d}} = (\alpha_i^{\mathrm{IID}})^2 - (\alpha_i^{\mathrm{OOD}})^2. \tag{11}$$

So we have:

$$\begin{aligned}
\xi_1^{\hat{d}+1} - \xi_1^{\hat{d}+1} = \sum_{i=1}^{\hat{d}} & \Bigg[ \left( \mathbb{E}^{\mathrm{OOD}}[\boldsymbol{x}_{\hat{d}} y]^\top \boldsymbol{v}_i^{\mathrm{OOD},\hat{d}} \right)^2 \left( \lambda_i^{\mathrm{OOD},\hat{d}} \right) \left( \frac{1}{\lambda_i^{\mathrm{ID},\hat{d}}} - \frac{1}{\lambda_i^{\mathrm{OOD},\hat{d}}} \right)^2 \\
& - \left( \mathbb{E}^{\mathrm{OOD}}[\boldsymbol{x}_{\hat{d}+1} y]^\top \boldsymbol{v}_i^{\mathrm{OOD},\hat{d}+1} \right)^2 \left( \lambda_i^{\mathrm{OOD},\hat{d}+1} \right) \left( \frac{1}{\lambda_i^{\mathrm{ID},\hat{d}+1}} - \frac{1}{\lambda_i^{\mathrm{OOD},\hat{d}+1}} \right)^2 \Bigg] \\
& + \left( \mathbb{E}^{\mathrm{OOD}}[\boldsymbol{x}_{\hat{d}+1} y]^\top \boldsymbol{v}_{\hat{d}+1}^{\mathrm{OOD},\hat{d}+1} \right)^2 \frac{((\alpha_{\hat{d}+1}^{\mathrm{ID}})^2 - (\alpha_{\hat{d}+1}^{\mathrm{OOD}})^2)^2}{(\lambda_{\hat{d}+1}^{\mathrm{ID},\hat{d}+1})^2 \, \lambda_{\hat{d}+1}^{\mathrm{OOD},\hat{d}+1}}.
\end{aligned} \tag{12}$$

From Eq. 10 and 12, we have the desired result. $\qquad\square$

