# ID and OOD Performance Are Sometimes Inversely Correlated on Real-world Datasets

# Appendix

## A Reviewers' FAQ

For transparency and to facilitate the reviewing process, we summarize questions and answers that arose during the review of an earlier version of this paper.

The most significant update to this version is the inclusion of **additional datasets**, and **experiments without a diversity-inducing method** (Section 5 and Appendix C). Some reviewers previously commented that our findings are not surprising, yet we continually see misunderstandings of their implications in the literature. We added a **discussion of such a recent case** [20] (April 2023) in Section 8. Other improvements include a discussion of **occurrences the existing literature** (Section 5.1) and various clarifications throughout the text.

**Q: Isn't the message of the paper unsurprising? Few would disagree that "focusing on ID performance alone may not lead to optimal OOD performance".**

**A:** We also would like this to be self-evident. Yet, multiple papers precisely conclude that focusing exclusively on ID performance is a fine strategy:

- "*If practitioners want to make the model more robust on OOD data, the main focus should be to improve the ID classification error*" [48]
- "*We see the following potential prescriptive outcomes: (...) the correlation between OOD and ID performance can simplify model development since we can focus on a single metric*" [24]

Many readers will clearly benefit from our exposition of the necessary caveats to such statements.

**Q: Past work on positive correlations is not a "for all" claim. Isn't it merely pointing out that this correlation is surprisingly strong in many benchmarks?**

**A:** This is indeed what the experiments show in "accuracy on the line" for example. But the takeaways (in this and other papers) are over-generalized and much overblown (see the citations in the previous answer above). The issue is clear when subsequent works apply this message uncritically and use the phenomenon ("accuracy on the line") as an unverified assumption on other datasets.

When this assumption serves to justify an experimental design (e.g. model selection) there is often no more chance to verify its validity later on, nor to recover from this faulty choice of assumptions (because they serve as premises for the whole analysis). The risk of such methodological mistakes in future research is why this paper is important.

**Q: Why the focus on the Camelyon17 dataset?**

**A:** We have now added experiments on five other datasets. They show that the phenomenon is not an isolated case.

Large-scale support for our message has also appeared after the release of a first version of this paper. Naganuma et al. [25] show that a re-evaluation of OOD benchmarks with a wider range of hyperparameters than previous studies leads to more diverse types of ID/OOD relations than the "linear trend".

Regarding the value of a large-scale evaluation of many datasets, note that this paper is in a very different situation from past studies claiming that positive correlations were widespread (cf. proof of existence vs. absence). The point of this paper is that "worst-case scenarios" (inverse correlation) are a possibility, hence the "best case" (positive correlation) cannot serve as an unverified assumption.

**Q: Why is the theoretical analysis restricted to linear models?**

**A:** Its value is precisely in demonstrating the phenomenon in a hypothesis space as simple as linear models. High-capacity models would be less surprising in displaying complex performance patterns.

**Q: Why aren't domain generalization (DG) methods investigated?**

**A:** Because the existence of predictors showing an inverse correlation is mostly relevant to the data and hypothesis space rather than learning methods. Using DG methods could have been a way to obtain a variety of models in the hypothesis space. We chose a less ad hoc solution with a general-purpose diversity-inducing method. This allows covering more of the ID/OOD spectrum than specific DG methods (i.e. we avoid the analysis to focus on the specific inductive biases of an arbitrary set of methods).

# B    Additional results on Camelyon17

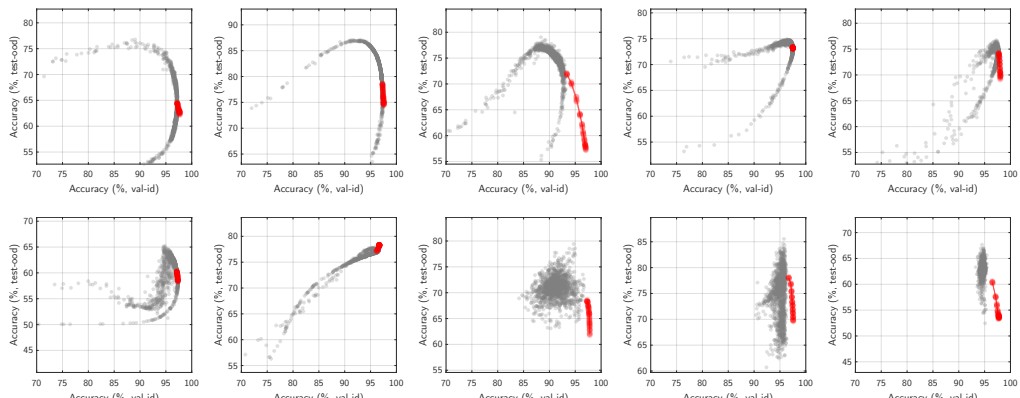

Figure 8: As in Figure 3, we show that higher OOD accuracy can be sometimes be traded off for a lower ID accuracy. Each panel shows results from a different pretrained model (i.e. pretrained with a different random seed). Each dot represents a linear classifier re-trained on features from this pretrained model with standard ERM (red dots ●) or with a diversity-inducing method [45] (gray dots ◉). The latter set includes models with higher OOD / lower ID accuracies.

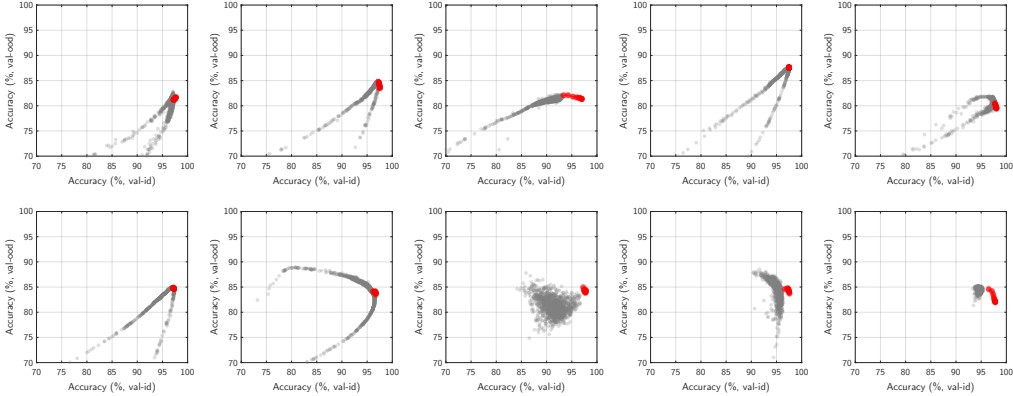

Figure 9: Same as in Figure 8, but using `val-ood` (instead of `test-ood`) as the OOD evaluation set.

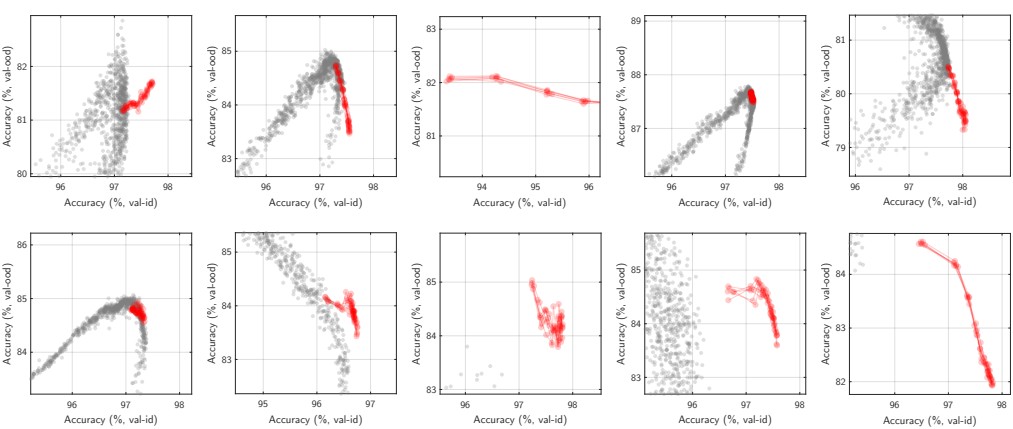

Figure 10: Same as in Figure 9, zoomed-in on ERM models (red dots ●).

## C Results on other datasets

In addition to Camelyon17, we performed experiments on five other datasets. We selected these datasets from the current literature on OOD generalization with no a priori knowledge of particular patterns of ID vs. OOD performance. We find inverse patterns to different extent on four out of five.