# OpenReview forum: "ID and OOD Performance Are Sometimes Inversely Correlated on Real-world Datasets"
_NeurIPS.cc/2023/Conference — NeurIPS 2023 spotlight_

### Official Review · Reviewer_zdij · 2023-06-27

**Soundness:** 3 good
**Presentation:** 4 excellent
**Contribution:** 3 good
**Rating:** 8
**Confidence:** 4

**Summary:**

This work performs an empirical study on the correlation between ID and OOD performances of models in vision and NLP. While some previous work claimed that there is usually a positive correlation between ID and OOD, this work argues that there are real datasets on which ID and OOD are negatively correlated, so ID should not be used as a proxy for OOD. The study was originally mainly done on Camelyon17, and this work also includes other datasets in Section 5, as well as similar observations in other papers. Section 6 provides a toy example where there is a trade-off between ID and OOD. Section 7 shows that the positive correlation occurs mostly in cases where the shift is mild, and with a larger shift the correlation could turn into a negative one. Section 8 points out several fallacies in previous work, including suggesting using the ID performance as a single metric, incorrect evaluation methodologies, etc.

**Strengths:**

- As someone very familiar with this area, I am well aware of the "accuracy-on-the-line" folklore that ID and OOD are positively correlated, and it has been reported in a lot of previous work. So to me, this submission is a breath of fresh air. A lot of work on OOD draws conclusions based on pure empirical observations. This is understandable because the OOD problem is an ill-posed problem by nature. However, the caveat is that lots of follow-up work takes these conclusions for granted, proposes new methods based on them, and claims to achieve "state of the art" while their evaluation is usually flawed. Such an unscientific methodology has haunted the OOD community for quite a long time, and I believe that this work conveys a strong message that this needs to be changed.
- This work is very clean and well written. The main argument of this work is by no means novel. In fact, in Section 5.1 the authors list previous papers that made similar arguments. Nevertheless, to my knowledge, this work is the first to convey this important message in such a clean way, and I believe that it can influence the perception and methodology of many researchers in the community.

**Weaknesses:**

- Section 6 is not very useful, and in my opinion, could be replaced with more experiments. It is very easy to construct a toy example where ID and OOD are negatively correlated. For example, if ID is a binary classification task, and OOD simply flips the labels, then of course they are negatively correlated. The reason why this "accuracy-on-the-line" has been there for so long is not because it is theoretically correct, but because it has been observed in many reports by different people. Everyone sufficiently familiar with OOD knows that there is no theoretical correlation between ID and OOD. I think this paper should be posed as a large-scale empirical study that shows that the correlation between ID and OOD on real datasets can be anything, so Section 6 is unnecessary.
- Section 5 is too short and the experimental results are not well presented. I suggest the authors remove Section 6 and put more experimental results in Section 5.

**Questions:**

- In Section 7, it is demonstrated that when the distribution shift is mild, the correlation between ID and OOD is usually positive. Are there any real datasets where the shift is mild but there is still a negative correlation between ID and OOD? In many real applications, it is safe to assume that the shift from train to test is mild. And if under this assumption ID and OOD is usually positively correlated, then one could argue that ID could still be used as a proxy for these applications. Note that I am not asking whether it is *possible* for there to be a negative correlation - It is easy to construct such a toy example. I am asking whether the correlation is always observed to be positive on real datasets with mild shift.

**Post rebuttal note:** I have read the rebuttal and had a discussion with the authors. I prefer to keep my score and vote for accepting this submission.

**Limitations:**

Limitations are discussed in Appendix A.

---

> ### Author Rebuttal · Authors · 2023-08-06
>
> Thanks for the thorough review. We are grateful for your appreciation of the importance of this paper for the OOD community.
>
> > Q1. Section 6 is not very useful
>
> We agree that the empirical results can stand on their own. But we are also fearful of readers dismissing the claims as "hand-wavy" unless a toy case is described in symbolic notation, as we do in Sec.6. We propose to move Sec.6 to the appendix, and instead expand Sec.5 to clarify the experimental details + include additional results currently in Appendix C.
>
> > Q2. In Section 7, it is demonstrated that when the distribution shift is mild, the correlation between ID and OOD is usually positive. Are there any real datasets where the shift is mild but there is still a negative correlation between ID and OOD?
>
> Very good question. We do not have such an example. We will however clarify in Sec.7 that this claim can only be guaranteed "*in the limit*" (i.i.d. setting).

---

> > ### Comment · Reviewer_zdij · 2023-08-10
> >
> > Thank you for the rebuttal. A good empirical paper does not need any theory, just like a good theory paper does not need any experiments. This used to be the common sense (just look at the NIPS papers 15 years ago), but unfortunately, lots of reviewers today do not know this. Thus, I would still suggest the authors remove Section 6 and add more experiments. To me, this paper is a good empirical paper with a clear and strong message, but can be made better with more experiments. And I also suggest the authors add a discussion on my question to the paper. This could be a possible limitation of this work.

---

> > > ### Author Response · Authors · 2023-08-15
> > >
> > > Completely agreed with this point of view. We are happy to follow these suggestions, which would indeed strengthen the message of the paper (moving the theoretical argument to the appendix, strengthening Sec.5 with additional results, and adding discussion about the above-mentioned question). Thanks for your time!

---

### Official Review · Reviewer_3Cei · 2023-07-03

**Soundness:** 4 excellent
**Presentation:** 4 excellent
**Contribution:** 3 good
**Rating:** 7
**Confidence:** 4

**Summary:**

The manuscript observe a phenomenon that ID and OOD performance can be inversely correlated because spurious correlations under the partially informative invariant feature (PIIF) assumption. The authors analyze the reasons of this missing phenomenon in previous studies, and conduct dataset-wise and shift-wise experiments. The paper provide a theoretical analysis and an OOD experiment guidance.

**Strengths:**

- The introduction of the ID/OOD inverse correlation is reasonable and easy-to-follow.
- The experiments are comprehensive.
- Theoretical analysis is equipped to provide insights.
- (Update) The empirical results coherently reflect previous theoretical findings, which is worth spreading in the OOD community.

**Weaknesses:**

~~The most obvious weakness of this paper is that the analyses are mostly experiment-centric, which leads to farraginous explanations and insights. This organization makes readers hard to follow the logics and insights of this paper. The authors may strive to highlight the inverse correlation is "surprising" instead of trying to explain this phenomenon more natural and intuitive.~~

The overall idea can be clearer and more insightful if the authors use causality as the backbone. For example, according to section 6, the observed phenomenon is **natural** with the structural causal model analysis under the partially informative invariant feature (PIIF) assumption.

Other weaknesses:
- Line 193: The structural causal equation definition is not formal. Use ":=" instead of "=".
- Line 102: Link is not valid.

~~Although I lean to accept, I'm not sure if this version of paper is enough beneficial for the research community.~~

The authors should revise their paper according to their proposal in the discussion period.



**Questions:**

I suggest extending section 6 and reorganizing the paper based on it.

**Limitations:**

No limitations, broader impacts, and licenses included.

---

> ### Author Rebuttal · Authors · 2023-08-06
>
> Thanks for your time and suggestions.
>
> > The most obvious weakness of this paper is that the analyses are mostly experiment-centric
>
> On the contrary, this is actually the key strength of the paper. The theoretical possibility of an inverse ID/OOD correlation is almost trivial (especially from a causal point of view, as rightly noted). The key strength of this paper is to show that this pattern is not only a worst-case pathological scenario, but does appear in multiple real-world datasets.
>
> The suggestion to emphasize the theoretical argument is completely at odds with reviewer zdij, who noted: "*It is very easy to construct a toy example where ID and OOD are negatively correlated.*", even recommending to remove Sec.6 and emphasize the paper "*as a large-scale empirical study that shows that the correlation between ID and OOD on real datasets can be anything*".

---

> > ### Comment · Reviewer_3Cei · 2023-08-10
> >
> > Thank you for your reply.
> >
> > I can consider updating my evaluations according to the current OOD circumstances as reviewer zdij mentioned.
> >
> > Although I acknowledge the contributions of this paper as an experimental work, I disagree with the suggestion of removing Sec. 6 by reviewer zdij. The `OOD problem is an ill-posed problem` only because we cannot achieve the second level of Judea Pearl's causality ladder learning on current observational distributions. In such scenarios, learning using ERM (only observational data) can lead to all uncontrolled cases from a theoretical view. This is why I mention this phenomenon is natural, and it also noticed by previous works as you mentioned in the paper. This is also the reason I'm against removing the theoretical analysis. The community should not be influenced by various case studies back and forth on an unspecified problem: using an observational learner (ERM) to solve OOD problems.
> >
> > *When there is no theory we can depend on, empirical validations are great and novel enough. However, when there are theories we can rely on, the community deserves a clearer and stronger claim.* Therefore,  I still suggest aligning the experiment results with current theoretical studies to make the manuscript clear and organized.

---

> > > ### Author Response · Authors · 2023-08-15
> > >
> > > Thanks for the thoughful review. We completely agree with the importance of established theories on the topic. Reviewers zdij & p2Jz seem only concerned about the alignment of Sec.6 with the experiments. Therefore, we propose to move it to the appendix (as they suggest) and include instead a stronger, shorter argument (as you suggest) reminding that OOD generalization is fundamentally underspecified when only rung-1/observational data is available.
> > >
> > > We propose to include this message upfront (in the introduction) to make it clear that it is a fundamental limitation of ERM. This should then clarify the role of the empirical results in the rest of the paper, and avoid future back-and-forths void of theoretical bases.

---

> > > > ### Comment · Reviewer_3Cei · 2023-08-16
> > > >
> > > > Thank you for your reply. After reconsideration, I agree that this paper has the important message worth sharing, and the proposal you mentioned is reasonable. Please revise the paper accordingly.
> > > >
> > > > Score update: $5\rightarrow 7$

---

### Official Review · Reviewer_yjiA · 2023-07-04

**Soundness:** 3 good
**Presentation:** 3 good
**Contribution:** 3 good
**Rating:** 6
**Confidence:** 4

**Summary:**

The paper presents an empirical examination of ID vs. OOD performance on several real-world datasets showing inverse correlation patterns can indeed arise in many datasets and experimental settings. This conflicts with past papers mainly observing a positive dependence structure. The paper further discusses why past studies missed such patterns and theoretically analyzes how inverse correlation patterns can arise.

**Strengths:**

- The paper highlights and important OOD generalization misconception, namely that increased model performance on in-distribution data is enough to achieve good OOD generalization. As the paper points out, it should be self-evident that this should not hold in general. Still, many works suggest that improving ID generalization helps OOD generalization. Model diversity seems to be a key factor that previous studies failed to account for. This is ensured via a training dynamics argument by examining model performance during a training run. I find this treatment convincing.
- The provided experimental evidence shows that inverse correlations are possible under starker distribution shifts and that previous results on positive correlations were mainly obtained by a restrictive experimental setting. It makes sense that an inverse trend should arise at some point under misspecification.
- The paper points out specific cases where the positive ID-OOD generalization insight has been observed but also where a negative relationship has been observed. This paper still seems to add value as in contextualizes all of these occurrences while explaining why a positive dependence structure might have appeared.
- I also like that the paper contains a theoretical section showing that an inverse correlation is expected under a spurious feature. At the same time, I am not sure it adds that much value as it mainly relies on a formalization of the effect of spurious correlations which have been previously theoretically analyzed in the domain generalization literature.

**Weaknesses:**

- Although the paper does provide meaningful value in analyzing the ID-OOD tradeoff, the paper sometimes mentions this idea as a new insight. I am not sure this is necessarily true. I would make sure not to oversell the "discovery of inverse ID-OOD tradeoffs" but not lose focus on explaining how previous papers could have overlooked this possibility (which the paper already does for the most part but not everywhere).
- I found point 4 in the description of Figure 2 confusing. The difference in the ID axis as compared with panel 2 does not appear to be large. 2 seems to show a more consistent variance while 4 seems to show a growing variance. Would it be appropriate to refer to theses as homoscedastic / heteroscedastic noise models, respectively?
- Last point in 3.1 (lines 182-183) is missing references; it would be good to back up this claim.

### Post-rebuttal

I have read the author's rebuttal and found that my concerns were adequately addressed. I maintain my positive score.

**Questions:**

- Why was the diversification method not used in the second set of experiments?

**Limitations:**

- Although limitations of the approach are not explicitly discussed, current shortcomings and future work are discussed in section 8 and 9.

---

> ### Author Rebuttal · Authors · 2023-08-06
>
> Thanks for your time and suggestions.
>
> > I would make sure not to oversell the "discovery of inverse ID-OOD tradeoffs" but not lose focus on explaining how previous papers could have overlooked this possibility (which the paper already does for the most part but not everywhere).
>
> This is a very valuable suggestion. We have edite the related statements throughout the paper to make this point unambiguous.
>
> > I found point 4 in the description of Figure 2 confusing. The difference in the ID axis as compared with panel 2 does not appear to be large. 2 seems to show a more consistent variance while 4 seems to show a growing variance. Would it be appropriate to refer to theses as homoscedastic / heteroscedastic noise models, respectively?
>
> The comment is valid for these particular examples (reproduced from [48]) but the "growing variance" in panel 4 is not a key characteristic (the data points could be all over the place). We added a note in the caption to make this clear.
>
>
> > L182-183 is missing references
>
> We added two supporting references about negative transfer in transfer learning:\
> [1] Characterizing and Avoiding Negative Transfer, Wang et al. CVPR 2019\
> [2] Loss-Balanced Task Weighting to Reduce Negative Transfer in Multi-Task Learning, Liu et al. AAAI 2019
>
>
> > Why was the diversification method not used in the second set of experiments?
>
> Because reviewers of an earlier version of the paper complained that the first set of experiments relied on the diversification method. We added this new set of experiments to show that diversity could be induced from simple, standard variations in hyperparameters and training methods.

---

> > ### Comment · Reviewer_yjiA · 2023-08-10
> > **Rebuttal Response**
> >
> > I thank the authors for their response to my comments. As a result, my concerns have been mostly addressed.
> >
> > > We added this new set of experiments to show that diversity could be induced from simple, standard variations in hyperparameters and training methods.
> >
> > It would have been great if the authors had included these experiments in the rebuttal PDF.
> >
> > ---
> > I leave my score unchanged.

---

### Official Review · Reviewer_Q2JA · 2023-07-07

**Soundness:** 3 good
**Presentation:** 3 good
**Contribution:** 3 good
**Rating:** 6
**Confidence:** 3

**Summary:**

Through extensive experiments, the authors showed that ID and OOD performances can inversely be correlated. The key ingredient is to track the performances throughout the training dynamics, instead of looking only at the end point.


**Strengths:**

By tracking the training process, instead of the final point, the authors gave a more comprehensive view on the matter. This methodology should be more widely adopted, in my opinion.

**Weaknesses:**

While Theorem 1 demonstrates this diverse performance phenomenon can happen even to simple linear models, how NN does so was not analyzed. I'd like to see the authors analyze a shallow neural net. There could be broader and deeper theoretical insights.

**Questions:**

Has the authors conducted an experiment for two or more OOD data performances, for a single set of models trained on an ID data, where the OOD data can arguably be ordered in a way that becomes gradually more OOD to the ID training data? For example, one can train on CIFAR 10, and track its OOD performances on some selected OOD classes from CIFAR 100, and do the same on SVHN as a further away OOD dataset.

I'm interested in the interplay between training dynamics and the more OOD axis.

---

> ### Author Rebuttal · Authors · 2023-08-06
>
> Thanks for your time and suggestions.
>
> > Q1. Theorem 1 demonstrates this diverse performance phenomenon can happen even to simple linear models, how NN does so was not analyzed. I'd like to see the authors analyze a shallow neural net.
>
> The claims of this paper are much more fundamental: they are not restricted nor dependent of particularities of neural networks. The connection with training dynamics of NNs [1] would be interesting for future work. A relevant analysis of a shallow neural net appeared recently [2].
>
> [1] Pitfalls of Simplicity Bias in Neural Networks, Shah et al. 2020\
> [2] Simplicity Bias in 1-Hidden Layer Neural Networks, Morwani et al. 2023
>
> > Has the authors conducted an experiment for two or more OOD data performances, for a single set of models trained on an ID data, where the OOD data can arguably be ordered in a way that becomes gradually more OOD to the ID training data?
>
> This would be interesting indeed. The challenge is to obtain multiple test sets where the distribution shift can be quantified. The suggested ones would be problematic because of the different classes.

---

> > ### Comment · Reviewer_Q2JA · 2023-08-16
> > **Not addressing the weakness**
> >
> > > The claims of this paper are much more fundamental: they are not restricted nor dependent of particularities of neural networks. The connection with training dynamics of NNs [1] would be interesting for future work. A relevant analysis of a shallow neural net appeared recently [2].
> >
> > I'm not suggesting the paper's claims are restricted. I merely mean that Theorem 1 is, under the linear regression setting. This creates a bigger discrepancy between Theorem 1 and your experiments which are based on neural nets. Your paper's primary contribution is the experimental studies. I find Theorem 1 peculiar in the paper flow. I'd recommend either move it to the Appendix or replace it with a neural net version, which are the models used in your experiments.
> >
> > > The challenge is to obtain multiple test sets where the distribution shift can be quantified. The suggested ones would be problematic because of the different classes.
> >
> > Will ID (SVHN) and OOD (gradually binarized SVHN, e.g. replace the [0, 1, ..., 255] scale to [0, 5, 10, 255] ) do it? It doesn't sound difficult as long as we look for it.

---

### Official Review · Reviewer_p2Jz · 2023-07-26

**Soundness:** 3 good
**Presentation:** 3 good
**Contribution:** 2 fair
**Rating:** 5
**Confidence:** 4

**Summary:**

The paper examines the previously proposed idea that in-distribution (ID) and out-of-distribution (OOD) performance of models are positively correlated. Through various experiments, the authors demonstrate that ID and OOD performance can in fact be inversely correlated. They also discuss how the previous work have missed this observation due to selection bias, and criticize the conclusions made by them. Moreover, a theoretical analysis on the linear case supplements the empirical findings regarding inverse correlation of ID and OOD performance.

**Strengths:**

Disclaimer: The paper is merely experimental as it is evident from its title. Before reading this report, I have not seen such activity in the OOD generalization literature; Note that benchmarks such as DomainBed [1] are fundamentally different as they do not provide suggestions for OOD generalization based on mere experiments. To understand OOD generalization, I believe the theoretical analysis must be the foundation, and ideally the results need to be backed with experiments. However, similar work have been approved in the literature; for example Wenzel et al. [2] which has motivated the empirical study of ID and OOD performance was published in NeurIPS 2022 despite making the false claims that are elaborated in this work. Therefore, I see the relevance and necessity of a rebuttal such as this work to correct the mistakes made by the previous work, and complement the existing literature. Only for this reason, I value the efforts made by the authors, and I recommend accepting the paper.

- The paper addresses important shortcomings of the previous work in methods and evaluation.

- The paper is well-written, and easy to understand for the audience who might be unfamiliar with the literature. The Reviewers’ FAQ in the appendix is very helpful.

- The empirical evaluation is comprehensive.

[1]  I. Gulrajani and D. Lopez-Paz, “In search of lost domain generalization,” 2020, arXiv:2007.01434.

[2] Florian Wenzel, Andrea Dittadi, Peter Vincent Gehler, Carl-Johann Simon-Gabriel, Max Horn, Dominik Zietlow, David Kernert, Chris Russell, Thomas Brox, Bernt Schiele, et al. Assaying out-of-distribution generalization in transfer learning. arXiv preprint arXiv:2207.09239, 2022.


**Weaknesses:**

- The premise of section 6 is to explain the inverse correlation witnessed by the experiments on the real-world datasets. To this end, the authors chose a linear model, and proved (Theorem 1) that adding a spurious feature decreases the OOD performance while it boosts the ID performance. It is unclear how this result corresponds to learning diverse models (section 3). In other words, how do the models learned in the experiments are related to adding a spurious feature to the feature set of a regression model? Such correspondence is necessary for section 6 to be part of a cohesive report.

- The work is missing experiments on synthetic data to connect its theory and the real-world experiments. It would have been ideal if the authors used the same experimental method to sample from ID/OOD performance of models fitted to the data generated by the linear model described in section 6. Ideally, the plots in Figure 7 must be reproducible with the toy model and synthetic data for different instantiations of the linear model.


**Questions:**

- To confidently recommend acceptance of this paper, I need to see the relevance of the theoretical analysis (section 6) to the experiments. In the previous section, I elaborated my recommendations to make this connection happen, and I am asking the authors to address my concern.

- Is there a solid conclusion/suggestion made by the paper apart from the presence of inverse correlations between ID and OOD performance? Apparently, the paper suggests that contrary to the conclusions made by previous work, ID performance can be irrelevant to OOD performance, however, the authors do not provide an alternative guideline for practitioners to make claims about OOD performance. Such discussion would be very helpful for the audience.

**Limitations:**

Mentioned in the previous parts.

---

> ### Author Rebuttal · Authors · 2023-08-06
>
> Thanks for your time and suggestions.
>
> > I need to see the relevance of the theoretical analysis (section 6) to the experiments. In the previous section, I elaborated my recommendations to make this connection happen, and I am asking the authors to address my concern.
>
> The theoretical possibility of an inverse ID/OOD correlation is quite direct as noted by reviewer zdij ("*It is very easy to construct a toy example where ID and OOD are negatively correlated.*"). Our theoretical analysis (section 6) shows how to construct such a toy example, in a (hopefully) didactic way. It shows how ID and OOD metrics of a model can diverge, as training progresses, or as one considers data with more and more spurious features.
>
> **The important open question was whether such cases are merely pathological or do appear in real-world data.** Prior work strongly argued for the former, but we show that the latter is actually correct. The key strength of this paper is therefore in the observations on actual data [*] and *not in a theoretical argument*. We propose to make this point much clearer in the text.
>
> [*] Simulations on synthetic data would only distract from this main message.
>
> > the authors do not provide an alternative guideline for practitioners to make claims about OOD performance
>
> The takeaway is precisely that simplistic guidelines do not apply universally. Sec.9 does include actionnable takeaways ("*Can we avoid inverse correlations*") and promising research directions ("*Model selection for OOD generalization*").

---

> > ### Comment · Reviewer_p2Jz · 2023-08-10
> > **Rebuttal Response**
> >
> > Thank you for your rebuttal and clarification.
> >
> > >The theoretical possibility of an inverse ID/OOD correlation is quite direct as noted by reviewer zdij.
> >
> > It indeed is. However, the theoretical analysis that is provided in Section 6 is neither complementing the experiments, nor expands our theoretical understanding. I agree with reviewer zdij that an experimental work does not necessarily need to have a theoretical contribution. However, if theory and experiments are present in one paper, they have to correspond to one another, which is not the case in this manuscript as I elaborated in my review. I suggest removing the theoretical contribution or taking it to the appendix in order to preserve cohesion.
> >
> > > (...) The key strength of this paper is therefore in the observations on actual data, and simulations on synthetic data would only distract from this main message.
> >
> > Conducting synthetic experiments corresponding to the toy model presented in section 6 was a suggestion to bridge the gap between section 6 and the rest of the work. If you view such synthetic experiments as distracting from the main message of the paper, then by the same argument section 6 is also distracting from the main message of the paper.
> >
> > Had the previous work been grounded theoretically, such false claims wouldn't have been made about positive correlation between ID and OOD performances. The main reason that I weakly suggest accepting this merely experimental work in OOD generalization, is the mistake in previous work that must be corrected. **Thus, I leave my score unchanged.**

---

> > > ### Author Response · Authors · 2023-08-15
> > >
> > > Thanks for the discussion. We're 100% onboard with these recommendations. The need to correct mistakes made in prior work is indeed the key reason for publishing this paper.
> > >
> > > We propose to follow the suggestion of moving the theoretical argument to the appendix, and strengthening the experimental section (Sec.5) with the results currently in the appendix, which better support the main message of the paper.

---

### Decision · Program_Chairs · 2023-09-21

**Decision:**

Accept (spotlight)

**Comment:**

All reviewers recognized that the paper makes a valuable empirical counterpoint to major narratives in the literature on robustness and the relationship between ID and OOD generalization on real-world distribution shifts. The literature review and experiments are comprehensive, and the paper's new results are well-contextualized with results that were previously reported.

Because the paper brings new empirical insights to a highly topical question, I am recommending it for a spotlight.